# NK cells eliminate Epstein-Barr virus bound to B cells through a specific antibody-mediated uptake

Elisenda Alari-Pahissa[1]*, Michelle Ataya[1], Ilias Moraitis[2], Miriam Campos-Ruiz[1], Mireia Altadill[1], Aura Muntasell[3,5], Anna Moles[4], Miguel López-Botet[1,5,6]*

1 University Pompeu Fabra, Barcelona, Spain, 2 University of Ioannina, Ioannina, Greece, 3 Autonomous University of Barcelona, Barcelona, Spain, 4 Department of Experimental Pathology, IIBB-CSIC, IDIBAPS, Barcelona, Spain, 5 Hospital del Mar Medical Research Institute (IMIM), Barcelona, Spain, 6 Immunology laboratory, Dpt. of Pathology, Hospital del Mar, Barcelona, Spain

* elisenda.alari@upf.edu (EAP); miguel.lopez-botet@upf.edu (MLB)

## Abstract

Epstein Barr virus (EBV) causes a highly prevalent and lifelong infection contributing to the development of some malignancies. In addition to the key role played by T cells in controlling this pathogen, NK cells mediate cytotoxicity and IFNγ production in response to EBV-infected B cells in lytic cycle, both directly and through antibody (Ab)-dependent activation. We recently described that EBV-specific Ab-dependent NK cell interaction with viral particles (VP) bound to B cells triggered degranulation and TNFα secretion but not B cell lysis nor IFNγ production. In this report we show that NK cell activation under these conditions reduced B cell transformation by EBV. NK cells eliminated VP from the surface of B cells through a specific and active process which required tyrosine kinase activation, actin polymerization and Ca2+, being independent of proteolysis and perforin. VP were displayed at the NK cell surface before being internalized and partially shuttled to early endosomes and lysosomes. VP transfer was encompassed by a trogocytosis process including the EBV receptor CD21, together with CD19 and CD20. Our study reveals a novel facet of the antibody-dependent NK cell mediated response to this viral infection.

## Author summary

Epstein-Barr virus (EBV) is a member of the herpesvirus family which causes a frequent and lifelong infection. The immune system is unable to fully eliminate the virus, which remains dormant in infected B lymphocytes. EBV reactivation leads to the production of new infective particles, spreading to other cells and favoring its transmission. EBV infection goes generally unnoticed in healthy individuals, though it may occasionally cause a disease termed Infectious Mononucleosis, as well as severe disorders in patients with a defective immune response. Remarkably, EBV has oncogenic potential contributing to the development of some tumors, and has been associated to autoimmune diseases. T lymphocytes and Natural Killer (NK) cells play an essential role in the defense against EBV, killing infected cells when the virus reactivates. Antiviral NK cell functions may be

**Funding:** The work was supported by grants SAF2016-80363-C2-1-R (MINECO/FEDER/UE), awarded to MLB by Ministerio de Economía y Competitividad and PID2019-110609RB-C21/AEI/ 10.13039/501100011033 awarded to MLB by Agencia Estatal de Investigación. The funders had no role in study design, data collection and analysis, decision to publish, or preparation of the manuscript.

**Competing interests:** The authors have declared that no competing interests exist.

also triggered by antibodies (Ab) recognizing infected cells. In this report we provide the first evidence supporting that NK cells in combination with anti-EBV Ab are able to eliminate the virus attached to the surface of B cells, reducing their infection without killing them.

## Introduction

Epstein-Barr virus (EBV) is a human γ herpesvirus which causes a highly prevalent and life-long persistent infection, generally asymptomatic in healthy individuals. Yet, primary infection may cause acute Infectious Mononucleosis, and viral reactivation in immunocompromised individuals may promote the development of lymphoproliferative disorders and hemophago-cytic lymphohistiocytosis (HLH). Moreover, EBV infection underlies the development of hematologic (i.e. Burkitt and Hodgkin lymphoma) and epithelial (i.e. nasopharyngeal carcinoma) tumors [1], being as well associated to some autoimmune disorders (e.g. Multiple Sclerosis) [2].

EBV transmission occurs mainly through saliva followed by infection of epithelial cells as well as tonsillar and adenoid naïve B cells. The interaction of the viral envelope glycoproteins gp350/220 with complement receptor 2 (CD21) is an important step in infection of B cells [3], although CD35 can also act as receptor [4]. B cells constitute the reservoir for virus latency, that may develop with different patterns [5]. Activation of latently infected B cells and differentiation to plasma cells activates the lytic cycle leading to the production of infective viral particles (VP) [6].

T cells are essential for the control of EBV infection and increasing evidence for a role of NK cells has been obtained [7,8]. Downregulation of HLA class I (HLA-I) molecules, together with expression of NKG2D and DNAM-1 ligands by EBV-infected cells in lytic cycle promoted NK cell cytotoxicity and IFNγ production [9]. Tonsillar CD56$^{bright}$ NK cells have been reported to play a role in the response to EBV [10,11] and experiments in humanized mice support the contribution of NK cells in control of the viral infection [12]. Observations in primary immunodeficiencies reveal that a variety of defects affecting the development and function of cytotoxic T and NK cells may increase susceptibility to EBV infection [1,5,13]. Recently, the first reported complete deficiency of FcγR-IIIA (CD16A) was associated to chronic EBV replication, supporting a contribution of antibody-dependent cell mediated cytotoxicity (ADCC) in immune defense against this pathogen [14]. Early studies showed that NK cells in combination with serum Abs specific for viral antigens displayed on the surface of infected cells (e.g. gp350/220) mediated ADCC against EBV infected cells in lytic cycle [15–17]. Compared to other activating NK cell receptors, FcγR-IIIA does not require additional costimulatory signals for triggering NK cell effector functions [18]. We reported that Ab-mediated NK cell response against EBV-infected B cells in lytic cycle triggered cytotoxicity as well as TNFα and IFNγ production. By contrast, when NK cells were confronted to VP-bound B cells, anti-EBV Abs triggered NK cell degranulation and TNFα secretion, but not target cell lysis nor IFNγ production [19]. In this report we provide evidence supporting that NK cells, activated under these conditions, uptake VP from the surface of B cells and internalize them through a specific and active process encompassed by trogocytosis. The mechanism of this novel facet of the NK cell mediated response to the viral infection has been characterized and the putative implications for the response to the pathogen are discussed.

## Results

### Interaction of NK cells with B cell-bound EBV particles in the presence of EBV seropositive sera (EBV S+) reduces transformation without inducing cell death

We previously showed that in the presence of EBV S+ NK cells degranulated and produced TNFα in response to B cells bound to viral particles (VP) derived from the human AKBM cell line, but did not mediate cytotoxicity and IFNγ production. Similar results were obtained with purified VP from the EBV-producing marmoset B95.8 cell line, commonly employed to infect B cells, promoting their transformation. In these experiments EBV S+ induced NK degranulation comparably to that triggered by an anti-CD20 mAb (Rituximab) dose of 12.5ng/ml but, in contrast, did not appreciably induce B cell death (Fig 1A and 1B). To explore the putative influence of NK cell activation on EBV B cell infection, we tested the effect of a short (4h) co-culture with NK cells and EBV S+ serum on B cell transformation, assessed as described in Materials & Methods. VP-bound B cells cultured alone with EBV S- or S+ sera showed similar transformation levels. By contrast, incubation with NK cells in the presence of EBV S+ significantly reduced the transformation rate as compared to that observed in the presence of EBV S- (Fig 1C and 1D). In these experiments, separation of B cells after the 4h co-culture ruled out delayed effects of activated NK cells. Moreover, EBV S+ was added after VP binding to B cells and, as mentioned above, did not inhibit their transformation in the absence of NK cells, thus excluding a neutralizing effect. These results suggested that a short interaction of NK cells and EBV S+ with VP-coated B cells might reduce their infection, independently of cytotoxicity.

### NK cells and EBV S+ specifically eliminate VP from B cells through an active process independent of proteases and perforin

To investigate the mechanism underlying the Ab-dependent NK cell reduction of EBV transformation, B cell-bound B95.8 VP were stained by indirect immunofluorescence with an anti-gp350 antibody before co-culturing them with NK cells. After co-culture, gp350 was analyzed on B cells gating them as shown in S1 Fig. Remarkably, gp350 staining was markedly reduced in the presence of EBV S+ and NK cells (Fig 2A). Similar results were obtained in experiments performed with B cells coated with AKBM-derived VP (Fig 2B). The effect was exclusively mediated by EBV S+ and restricted to the IgG fraction (S2A and S2B Fig). To rule out a possible artifact derived from gp350 staining, we took advantage of the fact that AKBM cells are transfected with a plasmid encoding for GFP under a lytic cycle gene promoter, thus producing tagged VP. The GFP signal of B cells bound to AKBM-derived VP vanished after co-culture with NK cells and EBV S+ (Fig 2C). The effect was also observed in B cells of whole PBMC, which had been incubated with AKBM SN, stained for gp350 and cultured with EBV S + serum (S3A Fig). Thus, we considered whether FcγR-expressing monocytes might also mediate the effect. Selective depletion of NK cells or monocytes from PBMC had little or no effect respectively on VP elimination from B cells, in contrast to the clear inhibition observed when both cell types were depleted (S3B Fig). Moreover, co-culture of coated B cells with purified monocytes and EBV S+ also eliminated VP, confirming their participation in the effect (S3C–S3F Fig). Altogether, these results show that NK cells and monocytes can eliminate VP adhered to B cells in the presence of EBV S+.

Experiments were conducted to characterize the mechanism underlying NK cell- mediated VP elimination. Incubation in the presence of the Src family and Abl tyrosine kinase inhibitor Dasatinib [20] (Fig 3A) prevented the loss of B cell-bound gp350. Moreover, pre-treatment with EDTA or with the more specific $Ca^{+2}$ chelator EGTA comparably inhibited VP

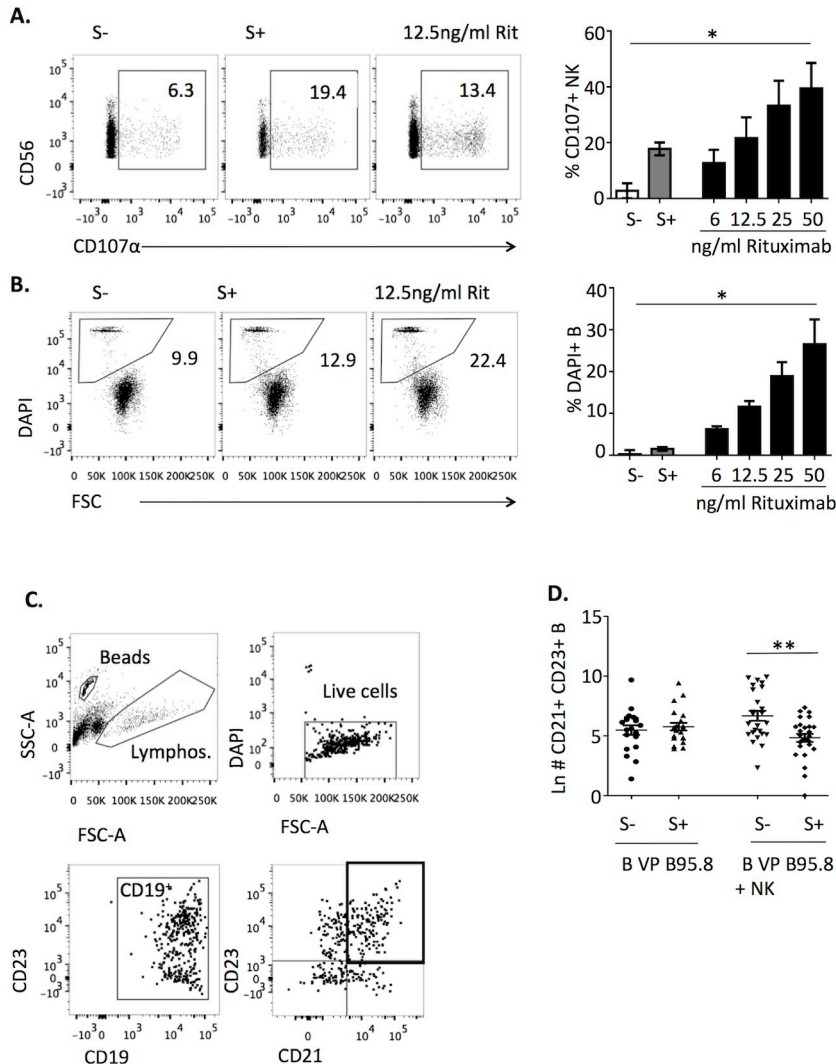

**Fig 1. Interaction of NK cells with B cell-bound EBV VP in the presence of EBV S+ reduces transformation without inducing B cell death.** (**A**, **B**) B cells were stained with fluorescent dye, coated with B95.8 VP for 1h at 4°C, washed, and cultured with autologous NK cells in the presence of EBV S-, EBV S+ serum or the indicated doses of Rituximab. After 4h, the percentages of CD107$\alpha^+$ NK cells (**A**) and of DAPI$^+$ B cells (**B**) were determined by flow cytometry. Background levels without B cells (for CD107$\alpha$) or without NK cells (for DAPI) were substracted. Plots of a representative experiment (left) and results of three (right) are shown. Statistical analysis was performed with Friedman test with Dunn's post-test. (**C**, **D**) B cells were incubated with B95.8 VP for 1h at 4°C, washed and cultured alone or with autologous NK cells in the presence of EBV S- or S+ serum for 4h. Subsequently, B cells separated from NK cells were plated at 50,000 cells /well, cultured for 2 weeks and stained for CD19, CD21, CD23 and DAPI. (**C**) Representative gating strategy. (**D**) Data are expressed as Ln of the number of live CD23$^+$ CD21$^+$ CD19$^+$ cells detected in 20–26 wells per condition from 3 experiments. Statistical analysis was performed with one-way ANOVA test with Bonferroni's multiple comparison test.

elimination from B cells (Fig 3A). These results indirectly supported that this effect involved an active process associated to CD16-triggered NK cell activation.

To explore whether perforin was involved in VP elimination, cells were treated with Concanamycin A or with the lysosomotropic agent Chloroquine, which increase the pH of lytic granules, promoting perforin degradation [21,22]. These agents effectively inhibited Rituximab-

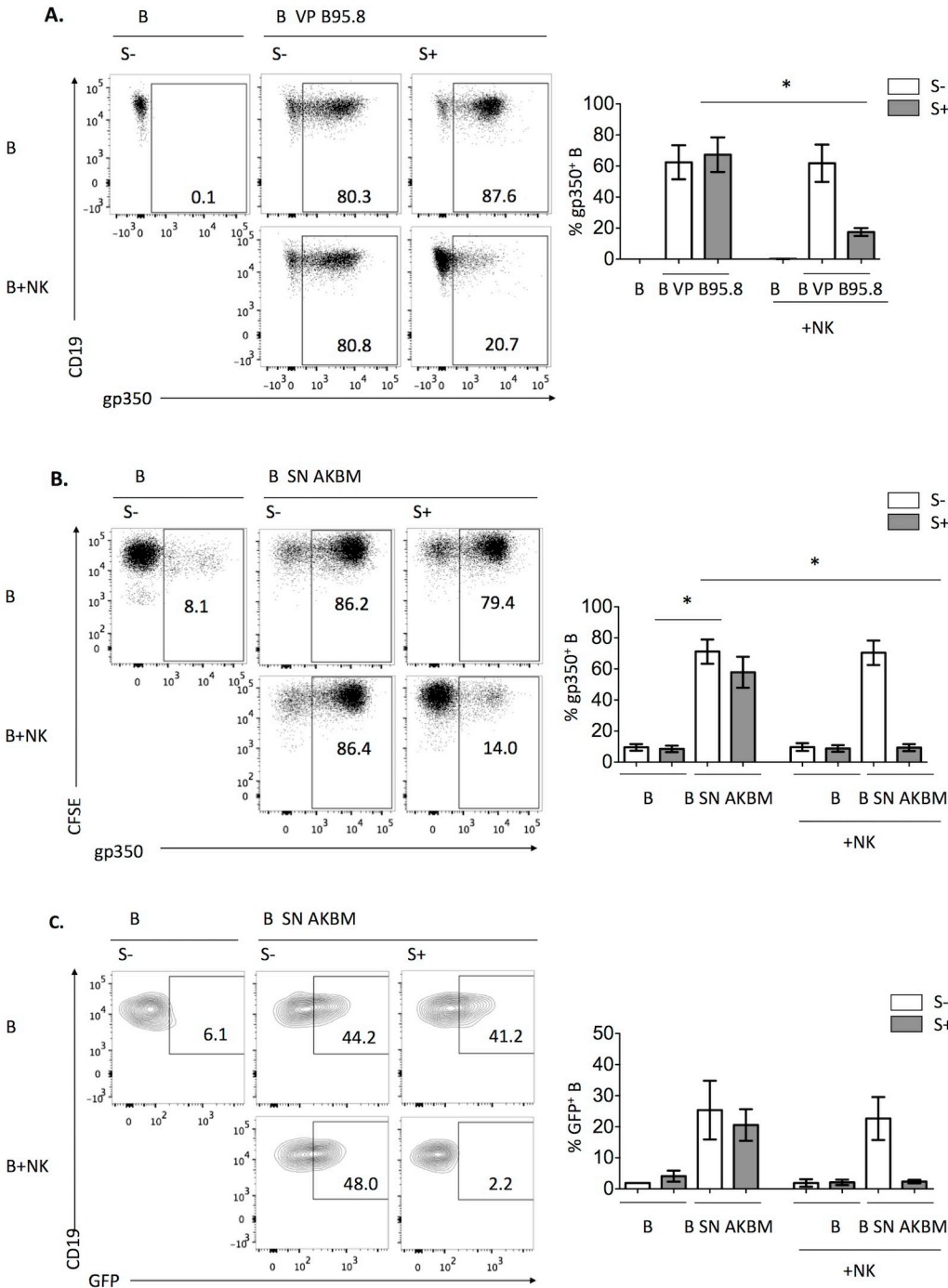

**Fig 2. NK cells in the presence of EBV+ serum remove VP attached to autologous B cells. (A)** B cells were pre-incubated with B95.8 VP for 1h at 4°C, washed, stained with anti-gp350 and cultured alone or with autologous NK cells in the presence of EBV S- or EBV S+ for 4h. The proportions of B cells (CD19$^+$) displaying gp350 were analyzed; representative plots (left) and results of four different experiments (right) are shown. **(B)** B cells were pre-incubated with AKBM SN, washed and stained with 0.3μM CFSE and for gp350. Samples were cultured alone or with NK cells in the presence of EBV S- or EBV S+ for 4h. The proportions of gp350$^+$ B cells (CFSE+) were analyzed; representative plots (left) and results of four experiments (right) are shown. **(C)** B cells were pre-incubated with AKBM SN for 1h at 4°C, washed, cultured with NK cells in the presence of EBV S- or EBV S+ for 4h and the proportions of GFP$^+$ B cells (CD19$^+$) were analyzed; representative plots (left) and results of three experiments are shown. **(A-C)** Statistical analysis was performed with Friedman test with Dunn's post-test.

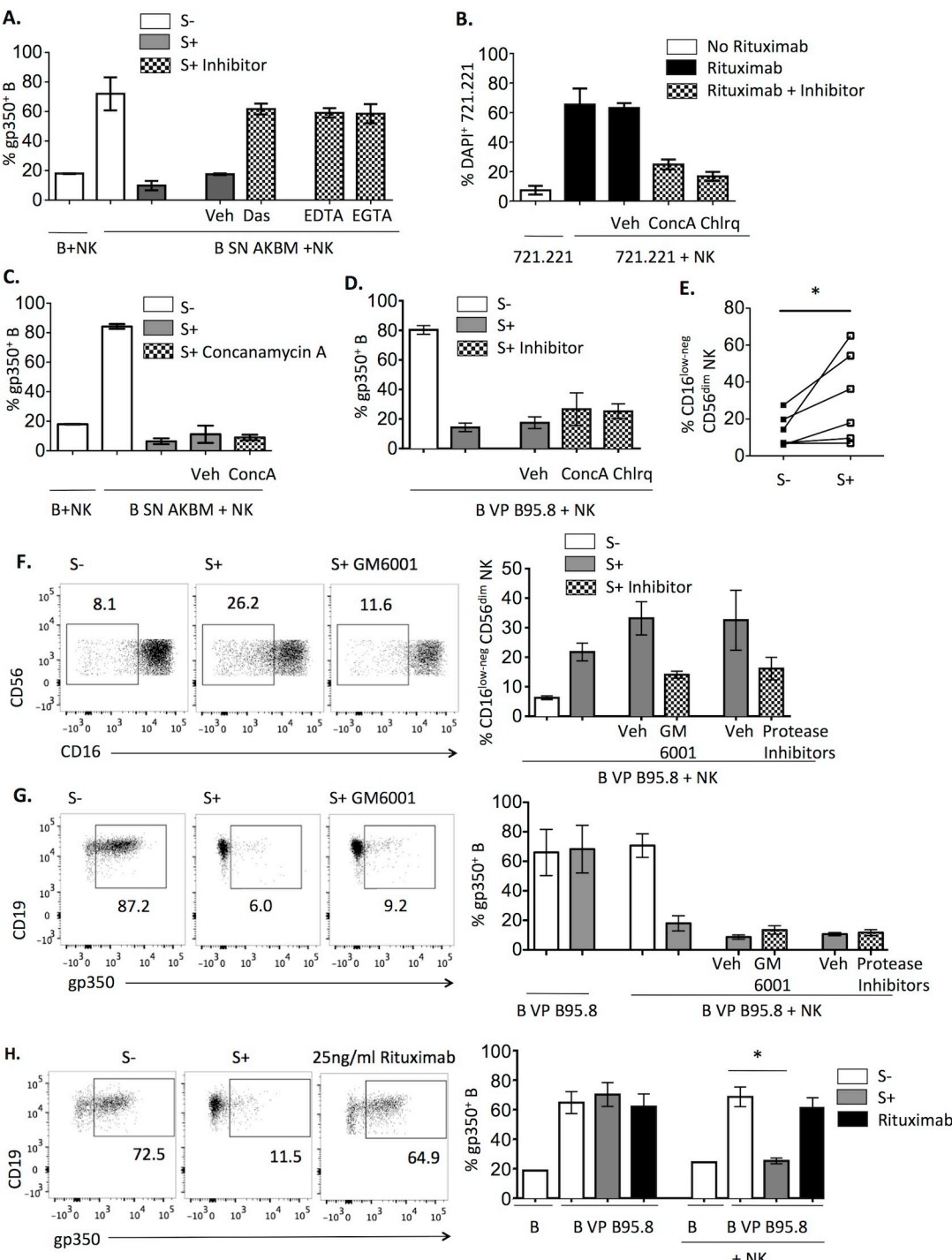

**Fig 3. EBV-specific Ab-dependent removal of B cell-bound VP by NK cells is an active process independent of perforin and proteases. (A, C-G)** B cells were pretreated with AKBM VP (**A, C, E**) or with B95.8 VP (**D, F, G**), stained after washing with anti-gp350 and cultured with NK cells together with EBV S- or EBV S+ sera for 4h. (**A**) NK cells were pre-incubated for 2h with 50nM Dasatinib or vehicle (DMSO), or for 30 min with EDTA (20mM) or EGTA (10mM) and the same concentrations were maintained during the co-culture. The proportions of gp350+ B cells detected in three (Dasatinib, EDTA) or two experiments (EGTA) are shown. (**B-D**) NK cells were pre-treated or not for 2h with 1μM of Concanamycin A, 50μM of Chloroquine or vehicle (DMSO) and co-cultured with 721.221 with or without 100ng/ml of Rituximab (**B**), or with AKBM VP- (**C**) or B95.8 VP-coated (**D**) B cells; the same concentrations of inhibitors were maintained during co-cultures. Proportions of DAPI+ 721.221 cells (**B**) or of gp350+ B cells (**C, D**) from four (**B**) or three experiments (**C, D**) are shown. (**E**) Proportions of CD16[low-neg] cells detected among CD56[dim] NK cells from six experiments are shown. Statistical analysis was performed with

Wilcoxon test. (**F-G**) NK cells were pre-treated or not for 2h with 25μM of GM6001, protease inhibitor cocktail of 200μM Leupeptin, 8μM Aprotinin, 15μM Pepstatin, 25μM GM6001 and 10μM Amastatin or the equivalent volume of vehicle (DMSO or $H_2O$ plus DMSO). B and NK cells were co-cultured for 4h in the presence of EBV S- or S+ sera, with or without fresh inhibitor or vehicle at the same concentration used for pre-treatment. Proportions of CD16[low-neg] cells within CD56[dim] NK cells (**F**) and of gp350[+] B cells (**G**) are shown. Representative plots of the indicated conditions (left) and results of four experiments (right) are shown. (**H**) B cells were incubated with B95.8 VP, stained for gp350 and cultured alone or with NK cells in the presence of EBV S-, S+ or Rituximab (25ng/ml) for 4h. The proportions of gp350[+] B cells were analyzed. Representative plots (left) and results of four experiments are shown. For statistical analysis Friedman test with Dunn's post-test was applied to the last 3 columns.

induced NK cytotoxicity against 721.221 targets in control experiments (Fig 3B) but did not affect VP loss (Fig 3C and 3D), thus excluding a role of perforin.

We hypothesized that VP elimination might involve proteases associated to NK cell activation. CD16A is shed by ADAM17 (also called MMP17 or TACE) [23,24] and MT6-MMP (also called MMP25) [25] metalloproteases. To address whether shedding of CD16 bound to anti-EBV IgG-VP complexes might detach them from B cells, experiments were carried out in the presence of the GM6001 metalloprotease inhibitor. CD16 downregulation induced in the presence of EBV S+ (Fig 3E) was reduced by GM6001 (Fig 3F); the inhibition was partial presumably reflecting that CD16 may be also internalized [24,26]. However, GM6001 did not affect VP elimination (Fig 3G), thus suggesting that the mechanism is independent from CD16 shedding by metalloproteases.

To address whether other proteases released by activated NK cells (e.g. granzymes) could mediate VP elimination, NK cell co-cultures with VP-bound B cells were performed in the presence of a mixture of inhibitors of serine- (leupeptin, aprotinin), cysteine- (leupeptin), acid- (pepstatin) and metallo-proteases (GM6001) as well as aminopeptidases (amastatin). Under these conditions, CD16 downregulation was confirmed to be inhibited (Fig 3F) without affecting VP elimination by NK cells (Fig 3G) further supporting that this process is independent of protease activity.

The possibility that VP elimination could take place as a bystander effect of NK cell degranulation induced by other B cell-targeting Abs was considered. Of note, anti-CD20 Rituximab at doses inducing a greater degranulation than EBV S+ (Fig 1A) did not promote VP elimination by NK cells (Fig 3H), indicating that the effect strictly depends on EBV-specific IgG.

## EBV S+ promotes transfer of VP from B cells to NK cells

Flow cytometry analysis revealed that gp350 was detected on NK cells following their co-culture with B95.8 VP-bound B cells in the presence of EBV S+ but not of Rituximab (Fig 4A). This observation was also confirmed in co-cultures with B cells treated with AKBM-derived SN in the presence of EBV S+ but not EBV S- from different individuals (Fig 4B).

To more precisely assess the relation between gp350 loss in B cells and its presence on NK cells, we compared by immunofluorescence the levels of gp350 detected on each cell population. In this analysis, the magnitude of the B cell-associated gp350 decrement was proportional to the increase detected in NK cells (Fig 4C). Consistent with these observations, gp350 MFI of the whole lymphocyte population did not significantly change (Fig 4D). Moreover, treatment with the metalloprotease inhibitor GM6001 or with the protease inhibitor mixture did not increase gp350 on NK cells (Fig 4E). Altogether, these data support that NK cells uptake and retain B cell-bound VP. Analysis of the kinetics of this process revealed that transfer of gp350 from B95.8 VP- (Fig 4F) or AKBM VP-bound B cells (Fig 4G) to NK cells was relatively rapid, as a substantial gp350 staining of NK cells was already detected 15' after adding EBV S+.

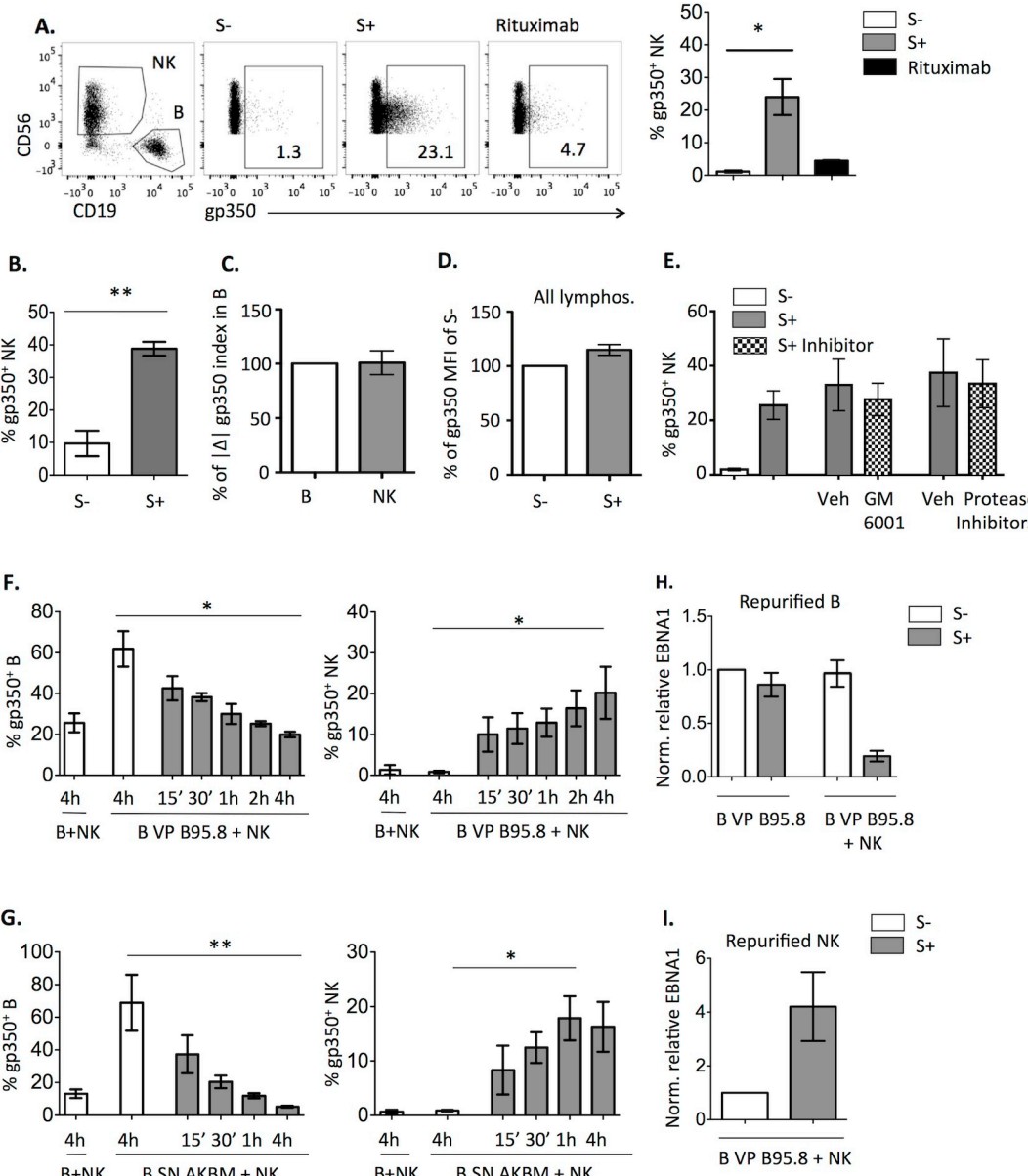

**Fig 4. EBV S+ induces VP transfer from B to NK cells. (A)** B cells were coated with B95.8 VP, stained for gp350 and incubated for 4h with NK cells in the presence of EBV S-, S+ serum or 25ng/ml of Rituximab. Proportions of gp350+ NK cells detected in representative plots (left) or four experiments (right) are shown. Statistical analysis with Friedman test and Dunn's post-test was applied. **(B)** B cells were incubated with AKBM SN, stained for gp350, co-cultured for 4h with NK cells in the presence of sera from various EBV S- (n = 5) or S+ (n = 6) donors, and analyzed for the proportions of gp350+ NK cells. **(C-D)** B cells were treated and co-cultured with NK cells as in (A), in the presence of EBV S- or S+ sera. **(C)** A gp350 fluorescence index was calculated for B and NK cells multiplying the percentage of the population by the corresponding gp350 MFI; the absolute variation ($|\Delta|$) of the index comparing conditions with EBV S- and S+ was calculated. For each experiment, the absolute variation of the index in NK cells was expressed as a percentage of that calculated for B cells. **(D)** For each experiment, MFI of all live cells in the co-culture with EBV S+ was expressed as a percentage of that calculated for EBV S- co-cultures. In **C-D** results of eight experiments are shown. **(E-G)** B cells were incubated with B95.8 VP **(E, F)** or AKBM VP **(G)**, stained for gp350 and co-cultured with NK cells, in the presence of EBV S- or S+ for 4h **(E)** or during the indicated times **(F, G)**. The proportions of gp350+ B **(F, G;** left) and NK cells **(E-G,** right) corresponding respectively to four **(E)** and three **(F, G)** experiments are shown. Statistical analysis was performed with Friedman test with Dunn's post-test. **(H-I)** B cells coated with B95.8 VP were cultured alone or with NK cells in the presence of EBV S- or S+ serum for 4h, followed by separation of B and NK cells. DNA was extracted from both cell fractions and analyzed by qPCR for the EBV gene EBNA1 and the endogenous gene 36B4. Relative quantification of EBNA1 in relation to 36B4 for B **(H)** and NK cells **(I)** was calculated and normalized to the first condition of each graph. Data correspond to three experiments.

VP released by EBV-infected cells have been reported to include gp350$^+$ vesicles together with virions [27]. Quantification of DNAase-resistant EBV DNA from B and NK cells, separated after the 4h co-culture, showed viral DNA loss from B cells (Fig 4H) and a concomitant gain by NK cells (Fig 4I), pointing out that virions were transferred to NK cells. The possibility that following this process NK cells might be infected by EBV was addressed, yet we did not detect viral gene expression when NK cells were re-purified and cultured alone for several days (S4 Fig). Altogether these results supported that anti-EBV specific Abs induced a rapid uptake by NK cells of VP bound to B cells without leading to detectable infection of NK cells.

## Cellular interactions involved in Ab-mediated NK cell activation and uptake of B cell-bound VP

To further characterize the EBV uptake process, experiments were carried out in the presence of Cytochalasin B, which blocks F-actin polymerization, inhibiting the formation of effector-target cell conjugates [28,29] and the resulting NK cell activation. As expected, this compound abolished NK cell degranulation, also preventing VP transfer to NK cells (Fig 5A). As Cytochalasin B does not directly inhibit NK cell degranulation [30,31], these results indicated that B cell-bound VP uptake involved cytoskeletal reorganization primarily promoted by their Ab-mediated interaction with NK cells.

To better characterize the EBV S+-induced transfer of VP from B to NK cells, confocal microscopy analysis was carried out following co-cultures at different time points. Consistent with flow cytometry observations, in the presence of EBV S- VP were almost exclusively found on B cells, whilst in co-cultures with EBV S+ VP fluorescence was predominantly detected on NK cells (Fig 5B and 5C). Different NK-B cell interaction patterns were observed, but VP were often detected close to or at the NK-B cell interphase (Fig 5B). To characterize the VP transfer dynamics, video microscopy analysis was performed. In co-cultures of NK cells with VP-bound B cells and EBV S+ (S1 Video, showing 41'), acquisition events could be observed which involved extensive NK-B cell contacts. However, no stable conjugate formation was observed and VP-bound B cells interacting with NK cells were not killed, confirming the main conclusions from static experiments. No VP transfer was detected with EBV S- (S2 Video, showing 41').

## NK cell uptake of B cell-bound VP entails trogocytosis of CD21 and other plasma membrane molecules

Experiments were conducted to investigate whether Ab-mediated VP-transfer might drag the EBV receptor. As shown in Fig 6A, CD21 was detected on the surface of NK cells following their interaction with VP-bound B cells. Remarkably, other B cell surface molecules including CD19, which forms a complex with CD21, and CD20 were also detected on NK cells (Fig 6B and 6C). As mAbs used to detect human CD21, CD19 or CD20 do not react with the B95.8 marmoset cell line, the possibility that these molecules could be on the VP envelope was ruled out. NK cells also acquired with a similar kinetics CD21 and CD19 from AKBM VP-coated B cells (Fig 6D and 6E).

These observations were reminiscent of trogocytosis [32–34], a process driven by receptor-ligand interactions which entails an active transfer of membrane patches, including the ligand of the driving receptor. To directly address this issue, B cells were stained with the PKH26 lypophilic dye before incubating them with B95.8 VP. In these experiments a small gain of PKH26 signal on NK cells was observed after co-culture with EBV S+ (Fig 6F). In comparison, Rituximab doses leading to similar degranulation induced greater levels of CD19 and PKH26-labeled membrane transfer (Fig 6G and 6H). These results suggested that NK cell

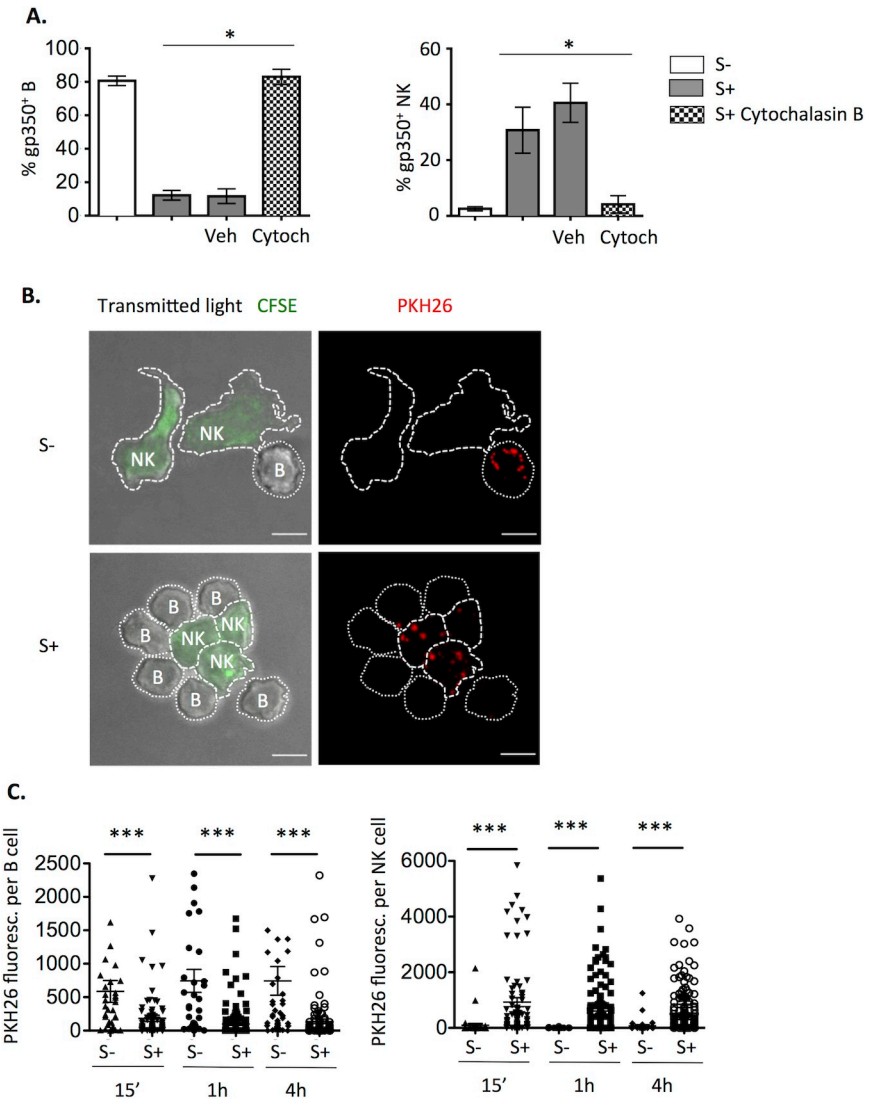

**Fig 5. Transfer of B cell bound-EBV particles to NK cells involves contact. (A)** B cells were coated with B95.8 VP, stained for gp350 and co-cultured with NK cells pre-treated for 2h with 20μg/ml of Cytochalasin B or the equivalent volume of vehicle (DMSO). Co-cultures were performed in the presence of EBV S- or S+ serum, with or without 20μg/ml of Cytochalasin B or vehicle for 4h. The proportions of gp350+ B (left) and NK cells (right) corresponding to four experiments are shown. Statistical analysis was performed with Friedman test with Dunn's multiple comparison test. **(B-C)** B cells were coated with PKH26-stained B95.8 VP, cultured with CFSE-stained NK cells and EBV S- or S+ for 15min, 1h or 4h, adhered onto coverslips and analyzed by confocal microscopy. **(A)** Images of Maximum Intensity of the stack of the EBV S- (top) or S+ (bottom) conditions at 15 min. Scale bar: 5μm. Data are representative of three experiments. **(B)** Quantification of total PKH26 fluorescence per B (left, 31–158 cells per condition) or NK cell (right, 34–116 cells per condition) from two different experiments. Statistical analysis was performed with Mann-Whitney U test.

uptake of VP is encompassed by limited trogocytosis events, indirectly induced by CD16A interaction with IgG-VP.

To assess whether CD21 transferred to NK cells might interact with EBV VP but avoiding the putative interference of B cell-bound VP involved with the Ab-mediated trogocytosis process, we took advantage of a report describing CD20 trogocytosis by NK cells upon ADCC against Rituximab treated B cells [35]. These observations were reproduced detecting also

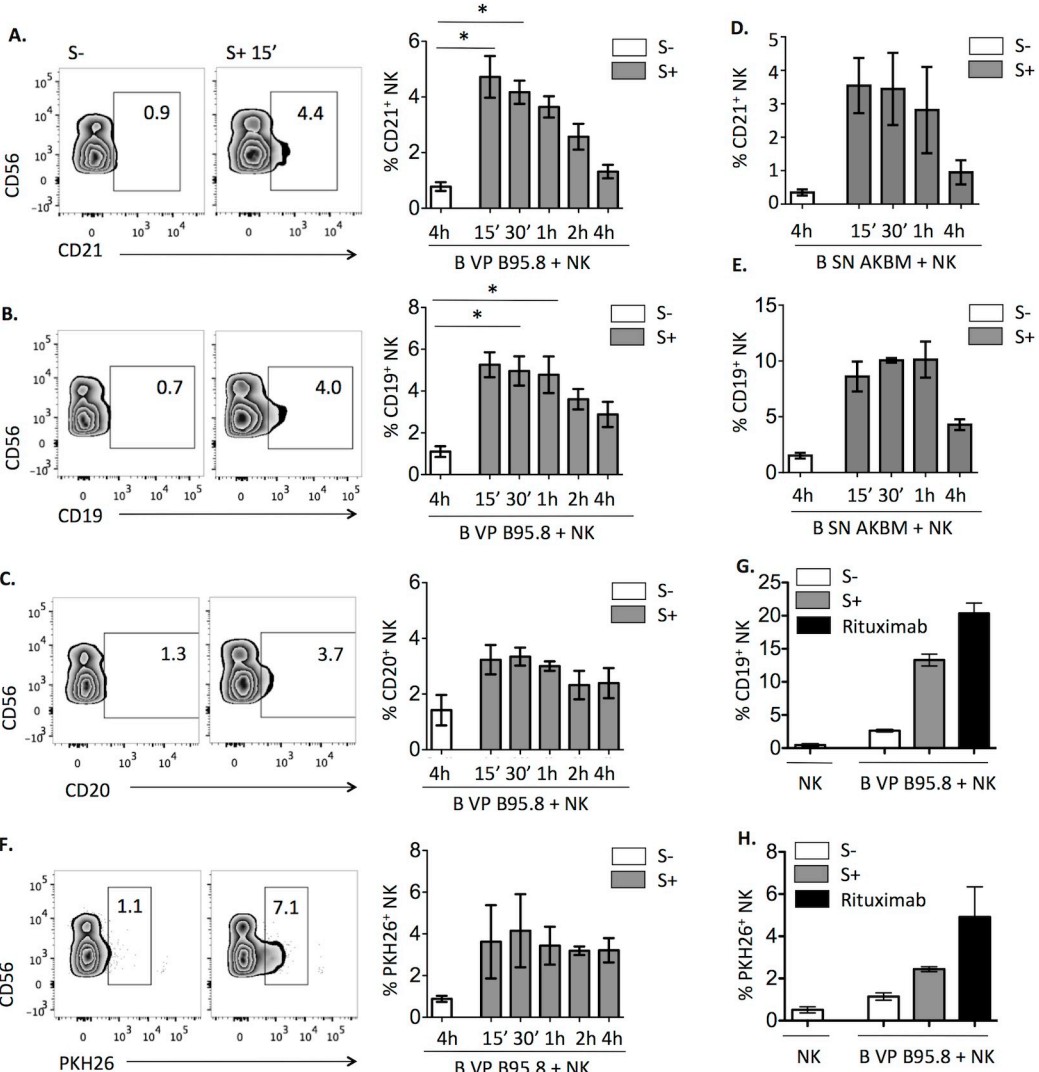

**Fig 6. Transfer of B cell-bound EBV particles to NK cells is encompassed by trogocytosis. (A-E)** B cells were coated with B95.8 VP (**A-C**) or AKBM VP (**D, E**), co-cultured with NK cells and EBV S- or S+ serum for the indicated times, and stained for CD21, CD19, CD20 and CD56. The proportions of CD21+ (**A, D**), CD19+ (**B, E**) and CD20+ (**C**) NK cells (gated as in Fig 4A) are shown. Representative plots of EBV S- at 4h and EBV S+ at 15min (left) and of four (**A-B**) and three (**C-E**) experiments (right) are displayed. Statistical analysis was performed with Friedman test with Dunn's multiple comparison test. (**F**) B cells were stained with 2μM PKH26, coated with B95.8 VP and co-cultured for the indicated times with NK cells and EBV S- or S+. Proportions of PKH26+ NK cells in representative plots of EBV S- at 4h and EBV S+ at 15min (left) and of three experiments are shown. (**G, H**) B cells were treated and co-cultured as in (**F**) but for 4h in the presence of EBV S-, EBV S+ or 12.5 ng/ml of Rituximab. The proportions of CD19+ (**E**) and PKH26+ (**F**) NK cells corresponding to two experiments are shown.

CD21 on NK cells (S5A Fig) at levels greater than those observed in our experimental system with VP-coated B cells, thus potentially increasing the sensitivity of the VP-binding assay. Under these conditions binding of PKH26-labeled VP was detectable on the surface of NK cells, and was partially blocked by anti-CD21 mAb (S5B and S5C Fig). These results indirectly suggest that CD21 transferred following NK cell activation by IgG-bound VP might bind new EBV VP; yet, the putative implications of this limited effect are uncertain.

## VP are internalized by NK cells and shuttled to early endosomes and lysosomes

We assessed whether NK cells internalized VP, evaluating as well VP entry into B cells, alone or co-cultured with NK cells in the presence of EBV S- or S+. To this end, B95.8 VP-coated B cells were incubated with anti-gp350 and stained with the PE-conjugated secondary Ab, before or after the culture (Fig 7A). A limited and relatively slow VP internalization by B cells was observed, leaving a substantial proportion of VP detectable on the B cell surface after 4h, and thus did not prevent the faster VP elimination seen when NK cells and EBV S+ were present (Fig 7B). In the latter setting, VP were transiently detected at the NK cell surface and gradually internalized (Fig 7C). In the same line, trypsin treatment after 4h-co-cultures with EBV S+ minimally reduced the gp350 signal detected in NK cells, consistent with its internalization (Fig 7D). Moreover, VP acquired by NK cells co-localized with Rab5 after 15 min and 1h of culture (Fig 7E) and, to a lesser extent, with CD107α after 1h and 4h (Fig 7F). Altogether, these results suggest that, following their uptake by NK cells, VP are internalized and transit, at least partially, to early endosomes and lysosomes.

## Discussion

In the present study we provide evidence of a novel antibody-dependent mechanism whereby NK cells can actively remove B cell-bound EBV particles, independently of protease and perforin activities. NK cell uptake of VP was followed by their internalization and, moreover, was associated to trogocytosis of B cell membrane molecules. NK cell incubation with VP-bound B cells in the presence of EBV S+ reduced B cell transformation, compared to that noticed with EBV S-. These observations indirectly supported that antibody-dependent NK cell elimination of B cell-bound EBV might partially prevent viral entry. As previously reported, NK cell activation in the same experimental system induced degranulation and TNFα secretion, in the absence of cytotoxicity and IFNγ production which play a key role in NK cell response to EBV-infected cells in lytic cycle. We previously hypothesized that NK cell activation under these conditions might transiently reduce their cytolytic granule content which, together with CD16 shedding, would transiently hamper their response to EBV-infected cells. Moreover, TNFα activation of the NFκB pathway might potentially enhance infection progression [36,37]. Consistent with previous observations in cytotoxicity assays, video microscopy analysis showed that B cells remained viable following their interaction with NK cells. These results indirectly supported that NK cell activation by IgG bound to VP on the B cell surface did not engage adhesion molecules (i.e. LFA-1 and CD2), which are key for synapse stabilization, polarized degranulation and optimal perforin insertion on the target cell plasma membrane [38]. Altogether these observations suggest that the same process might have opposite effects at different stages of EBV infection, preventing initial viral internalization in B cells but, on the other hand, potentially reducing the Ab-dependent NK cell response to EBV-infected cells. Dynamics of individual NK cell interactions with B cells bound to several VP indicated that the efficiency of this mechanism is limited, suggesting that its ability to reduce EBV B cell infection in vivo would ultimately depend on the relative proportions of NK cells interacting with B cell-bound VP in tissues. It is plausible that a similar process might also operate against other viral infections. Yet, in every case, the putative effectiveness to prevent infection would be predictably conditioned by different factors, particularly the specific mechanism and kinetics of VP internalization. In addition, the likelihood of NK and VP-bound B cells interactions in different tissues is important. In this regard the lymphoepithelial layer of tonsils and adenoids constitutes the main site of initial EBV replication [6,39–41] and of infection of naïve B cells. These migrate from high endothelial venules to the area beneath the crypt epithelium

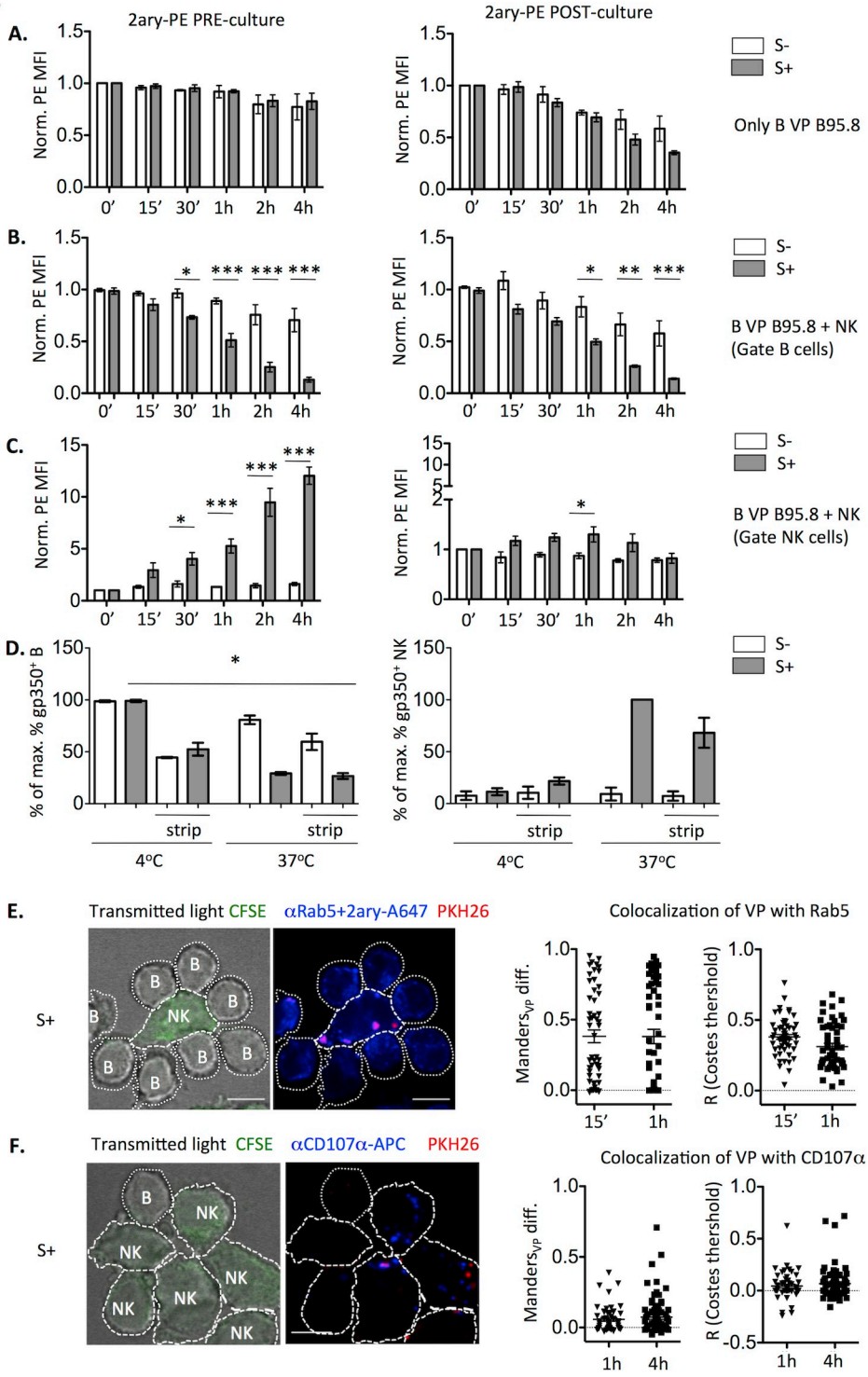

**Fig 7. EBV particles are internalized by NK cells and co-localize with endosome and lysosome markers. (A-C)** B cells were coated with B95.8 VP and stained by indirect immunofluorescence with anti-gp350 Ab, followed by the PE-conjugated secondary Ab either before (left) or after (right) being incubated alone (**A**) or with NK cells (**B, C**) in the presence of EBV S- or S+ for the indicated times. PE MFI was normalized to that detected on B cells alone (**A, B**) or NK cells (**C**) at time 0; data corresponding to three experiments is shown. Statistical analysis was performed with repeated measures mixed model ANOVA with Bonferroni post-test. (**D**) B cells were coated with B95.8 VP, stained for gp350, mixed with NK cells in the presence of EBV S- or S+ sera, incubated at 4°C or at 37°C for 4h, and treated or not with trypsin for 30 min. After trypsin neutralization, cells were washed and stained for CD19 detection. Since CD56

was eliminated by trypsin, NK cells were gated as CD19⁻ cells. The percentage of maximum gp350⁺ B (left) or NK cells (right) of each experiment is shown. Data correspond to three experiments. Statistical analysis was performed with Friedman test with Dunn's post-test. **(E-F)** B cells were coated with PKH26-stained B95.8 VP, cultured with CFSE-stained NK cells and EBV S+ for 15min, 1h or 4h, adhered onto coverslips, stained for Rab5 (**E**) or CD107α (**F**), and analyzed by confocal microscopy. Left: Representative slices at 1h (**E**) and 4h (**F**). Right: Corrected Manders coefficient and Pearsons coefficient (R) of co-localization of VP with Rab5 at 15 min and 1h (**E**) or with CD107α at 1h and 4h (**F**) within VP⁺ NK cells are shown. Data correspond to 51–78 cells per condition from two experiments. Scale bar: 5μm.

[6,41], where NK cells are found [42] and CD16 expression may be induced upon IL-2 activation [43]. On the other hand, other cell types expressing FcγR (e.g. CD16⁺ T cells, monocytes, dendritic cells, neutrophils) might also contribute to IgG-mediated elimination of B cell-bound VP. Actually, our preliminary data indicate that a similar effect may be mediated by monocytes. Further studies are required to explore the underlying mechanisms and specific functional implications for this lineage, including the role of different FcγR (i.e. CD16A, CD32 and CD64).

Following their uptake by NK cells VP were, at least partially, internalized and shuttled to early endosomes and lysosomes. HLA class II molecules are expressed by human activated NK cells, being detectable in a subset of circulating adaptive NK cells. We previously reported that activation by HCMV in the presence of specific antibodies promoted HLA class II expression in NK cells, which stimulated memory CD4⁺ T cells from HCMV+ donors in a HLA-II-dependent manner [44]. Thus, it is plausible that EBV uptake by NK cells might also promote specific Ag presentation to CD4+ T cells.

Uptake of B cell-bound VP by NK cells was encompassed by transfer of CD21 and other membrane molecules. This process was reminiscent of trogocytosis, defined as an active receptor-induced and contact-dependent intercellular transfer of membrane patches, containing the driving ligand(s) together with additional molecules [32–34]. NK cell-mediated trogocytosis has been observed upon direct formation of an activating immune synapse with target cells [45]. A similar process has been reported following the interaction of IgG bound to target cell surface antigens with FcγR-bearing effector cells (i.e. monocytes, macrophages, neutrophils and NK cells) [46]. In our experimental system, detection on NK cells of CD21 and CD19 suggested that the pull out was primarily mediated through B cell-bound VP, indirectly dragging other plasma membrane molecules (e.g. CD20). The transfer of B cell membrane molecules was limited as compared to that induced by Rituximab suggesting that the mechanical forces exerted indirectly through VP-IgG are weaker than those directly acting on the cell membrane.

Functional implications of trogocytosis have been reported depending on the cell types and molecules involved [47]; yet, information on this process in the context of viral infections is scarce. In this regard, antibody-dependent uptake of HIV bound to target cells mediated by a monocytic cell line was described [48]. In addition, NK cells were reported to acquire CD21 expression upon interaction with HLA-I defective B cell lines [45], potentially allowing NK cell infection by EBV. Our preliminary data indirectly suggest that CD21 acquired by NK cells through trogocytosis might bind EBV VP, further exposing them to a direct interaction with the pathogen.

In this regard, the possibility that EBV may infect NK cells is supported by the identification of rare EBV⁺ NK cell lymphoproliferative disorders [49], together with a report on in vitro EBV infection of NK cells [50], in which expression of three EBV genes (BZLF1, BALF2 and EBNA1) was detected in exposed NK cell lines. Based on this approach we addressed whether CD16-dependent internalization of trogocytosed EBV VP by NK cells might promote their

infection. However, no expression of the mentioned EBV genes nor of EBNA2 was detected in NK cells from co-cultures with VP coated B cells and EBV S+, in contrast to control EBV-exposed B cells. These results do not support that the Ab-induced EBV uptake may facilitate NK cell infection by EBV, but do not rule out that this might exceptionally occur in vivo.

A relevant role of CD16A in the control of EBV infection is supported by chronic viral replication observed in the first reported complete deficiency of this receptor, which impaired ADCC [14]. Another CD16A mutation, leading to deficient CD2 expression and natural cytotoxicity but unaltered ADCC, was associated to different viral infections, including EBV in some patients [51], in line with the role of direct NK cell recognition in viral infection control.

Our results suggest that NK cell-mediated elimination of B cell-bound VP might contribute to immune defense against EBV, complementing the role of CD16A-triggered ADCC and IFNγ production against infected cells in lytic cycle.

## Materials & methods

### Ethics statement

The study protocol was approved by the institutional Ethics Committee (Parc de Salut Mar CEIC 2018/7873/I).

### Cell lines and cultures

The AKBM human cell line, kindly provided by Prof. Martin Rowe, was generated by stable transfection of the EBV$^+$ Burkitt lymphoma cell line (Akata) with the pHEBO-BMRF1p-rCD2/GFP reporter plasmid, expressing GFP when the virus enters the lytic cycle [52]. The B95.8 is an EBV-producing marmoset B-lymphoblastoid cell that stably produces EBV VP. The HLA-I deficient human B-lymphoblastoid cell line 721.221 was used in standard NK cell cytotoxicity at a 4:1 effector:target ratio. All cultures were performed at 37˚C and 5% $CO_2$ in RPMI 1640 GlutaMax (Thermo Fisher Scientific) supplemented with penicillin (100 U/ml), streptomycin (100 μg/ml), sodium pyruvate (1 mM), and 10% FBS.

### Viral particle production and purification

To prepare viral particles from AKBM cells, they were incubated with 10μg/ml of F(ab')2 goat anti-human IgG (Cappel, Malvern, PA) at $2x10^6$ cells/ml for 2h to induce the lytic cycle, washed and further incubated at the same concentration in complete RPMI for 24h. Subsequently cultures were centrifuged at 350g for 10 min and cell free supernatants were aliquoted and stored at -20˚C.

B95.8 cells were cultured at $10^6$ cells/ml in complete RPMI for 3 days. Supernatants were collected after a centrifugation at 350g for 10 min, pre-cleared by centrifugation at 2000g for 30min, passed through a 0.45μm filter and EBV VP were purified as described [53] with some variations. Briefly, 10 ml of pre-cleared supernatants were ultracentrifuged over a cushion of 1ml of 50% iodixanol (Optiprep, StemCell) at 160,000g for 2h at 4˚C in a SW40Ti rotor in a XPN100 ultracentrifuge (Beckman Coulter). Nine ml of the upper phase were discarded and the 1ml left was mixed with the lower phase and layed on a iodixanol continuous gradient (27.5% - 50%) and ultracentrifuged at 160,000g for 2h at 4˚C in the same rotor. One ml fractions were taken and tested for the capacity to bind to B cells in PBMC samples by gp350 staining. Positive fractions were tested for DNaseI (New England Biolabs)-resistant EBV DNA and the DNA was purified and quantified by real time PCR. Fractions were used fresh for transformation experiments or aliquoted and frozen at -80˚C for later use. For PKH26 labeling of VP, B95.8 were washed and cultured as described above in exosome-free complete RPMI medium.

To prepare this medium, FBS was filtered through a 0.22 μm filter, ultracentrifuged at 120,000g for 18h at 4˚C, and the supernatant was passed through a 0.1μm filter. Penicillin, streptomycin and sodium pyruvate were also passed through 0.1μm filter. The SN was pre-cleared as described and VP were pelleted by ultracentrifugation at 50,000g for 1h at 4˚C, resuspended in 11ml of PBS and pelleted again at 50,000g for 1h at 4˚C for washing, resuspended in 0.5ml of 2μM PKH26 solution for 3min at RT. After adding 8 ml of exosome-free complete RPMI, samples were underlayed with 2ml of 20% iodixanol and ultracentrifuged at 190,000g for 2h at 4˚C. The upper layer and the interphase were removed and the lower phase was checked to label B cells of PBMC, aliquoted and stored at -80˚C. Negative controls were performed by processing a 2μM PKH26 dilution in the absence of VP, which did not stain B cells.

## Primary cell isolation and co-cultures

For most experiments, buffy coat samples were obtained from anonymized blood donors (Blood and Tissue Bank, Barcelona, Spain). For specific experiments, heparinized blood and serum samples were obtained from volunteer healthy individuals after written informed consent. PBMCs were separated on Ficoll-Hypaque gradient (Lymphoprep; Axis-Shield, Oslo, Norway). In some experiments PBMC were depleted of monocytes with Human CD14 MicroBeads (Milteny Biotec) and the positive monocyte fraction was used in co-cultures with B cells. NK cells were depleted with anti-CD56 MAb (Clone C128) and Sheep anti-Mouse IgG Dynabeads (Invitrogen).

B cells were negatively purified with the Pan-B Cell Enrichment Kit (StemCell). For coating with VP, B cells were incubated at 3-5x10^6 cells/ml with AKBM SN or, except for transformation experiments, at 10^6 cells/ml with a 1:1 mix of purified fraction from B95.8 SN with complete RPMI for 1h at 4˚C, and washed. Where indicated, samples were stained with anti-gp350 antibody (clone 72A1 Merck-Millipore) followed by a R-PE-conjugated secondary goat anti-mouse IgG+M F(ab')$_2$ (Jackson Immunoresearch) prior to co-culture. For membrane transfer experiments, B cells were stained with 2μM PKH26 (Sigma-Aldrich) prior to coating. In other experiments, AKBM SN-coated B cells were stained with 0.3μM CFSE (Molecular Probes, Invitrogen, Thermo Fisher) prior to staining for gp350.

For NK cell isolation, purified PBMCs were kept ON at 1.5x10^6 cells/ml with 200U/ml IL-2 and NK cells were purified with the NK Cell Isolation Kit (Miltenyi Biotech). NK cells were co-cultured with AKBM VP-coated B cells at a 2.5:1 ratio, or with B95.8 VP-coated B cells at a 1:1 ratio, in U bottom 96 well plates, in the presence of 3% EBV S-, 3% EBV S+ or Rituximab at the indicated doses, during 4h or the indicated times at 37˚C. Cells were stained for flow cytometry analysis of CD19, CD56, DAPI, and other markers where indicated. B cells and NK cells were gated as shown in Figs 4A and S1. Doublets were excluded based on SSC-A vs SSC-H distribution and live cells were gated as DAPI$^-$ cells. In some experiments, cells were incubated in Trypsin-EDTA solution (Sigma) for 20 min at 37˚C and washed in complete RPMI before staining. Anti-CD21 MAb (clone B-E5, mouse IgG2a; Abcam) or isotype control (anti HLA class I clone W6/32) were used in blocking experiments at a final concentration of 20μg/ml. Where indicated, after co-culture, various wells of the same condition were pooled and B cells and NK cells were re-purified with the Pan-B Cell Enrichment Kit (StemCell), collecting the negative and positive fractions, respectively, and analyzed for EBV DNA. To assess whether EBV might infect NK cells, these were re-purified with Dynabeads Untouched Human NK cells Kit (Invitrogen, Thermo Fisher) after co-culture with EBV-infected B cells in the presence of EBV S+ serum, and further incubated at 50,000 cells/well for 2, 5 or 15 days in flat bottom 96 well plates in the presence of 200U/ml IL-2 and analyzed for EBV RNA. For

cytotoxicity experiments, B cells were stained with 2μM PKH26 prior to VP-coating, and after 4h co-culture with NK cells, DAPI was added directly to culture media, incubated for 10 min on ice and DAPI$^+$ B cells (PKH26$^+$) were analyzed by flow cytometry. For degranulation assessment, co-cultures were performed in the presence of anti-CD107α-APC (Clone H4A3, BD Fastimmune) and 5μg/ml Monensin, stained, and CD107α$^+$ NK cells (CD56$^+$) were analyzed by flow cytometry.

Assessment of B cell transformation was performed as described [11] with variations. B cells were washed with PBS and incubated with ~ 0.75x10$^6$ EBV copies per 10$^6$ B cells in a mix of 80% volume of B95.8 purified SN fraction (in 25% Iodixanol) and 20% volume of HBSS 2% FCS for 1h at 4˚C. Subsequently, they were washed extensively, co-cultured at 1:1 ratio with NK cells and 3% EBV S- or S+ sera for 4h, re-isolated with the Pan-B Cell Enrichment Kit (StemCell), plated at 50,000 B cells/well in U bottom 96 well plates in 200μl of complete media and cultured for 2 weeks, during which the cultures were split twice. Individual well samples were stained for CD19, CD21, CD23 and DAPI and analysed by flow cytometry; 10,000 Blank Calibration Particles (BD Biosciences) were added to each cytometer before analysis. The numbers of live CD19$^+$ CD21$^+$ CD23$^+$ cells were recorded.

## Reagents

Leupeptin and Aprotinin (Sigma) were reconstituted in aqueous solution and stored at -20˚C. Pepstatin, Chloroquine (Sigma), Amastatin, GM6001 (Enzo Life Sciences), Dasatininb (Selleck Chemicals), Concanamycin A (Santa Cruz Biotechnology) and Cytochalasin B (ChemCruz) were reconstituted in DMSO and stored at -20˚C.

## Serum samples and IgG purification

Serum samples were collected, heat-inactivated (56˚C, 30 min), stored at −20˚C and analyzed for IgG Abs against EBNA1 and VCA EBV antigens with standard clinical diagnostic tests (*Laboratori de Referència de Catalunya*, Barcelona). Blood donors were considered seronegative for EBV when the test was negative for both EBV antigens. Where indicated, IgG was purified with Protein G-Sepharose (GE Healthcare) following standard protocols, and the negative fraction was also stored.

## Flow cytometry

The antibodies used were CD19-PECy5 (HIB19, Biolegend), CD56-APC-Cy7 (clone HCD56, Biolegend), CD21-FITC (Clone B-ly4, BD Pharmingen), CD23-PE (Clone EBV CS-5, Biolegend), CD20-BV510 (Clone 2H7, BD Horizon), CD16-PECy7 (Clone 3G8, BD) and CD14-APC (Clone 61D3, Invitrogen). Cell samples were washed once with PBS. Antibodies for cell surface markers and DAPI (final concentration of 2 μg/ml, Sigma-Aldrich) were employed diluted in PBS with 10μg/ml human aggregated IgG, 2% FBS and 2mM EDTA. Cells were incubated with Abs for 15min at 4˚C, washed twice with PBS, resuspended in PBS with 2% FBS and 2mM EDTA and analyzed in an LSR II flow cytometer (BD Biosciences). Data were processed with FlowJo X software (TreeStar).

## EBV DNA and RNA purification and real time PCR

For quantification of EBV genome copies in B95.8 VP purified preparations, DNA from DNase-treated samples was purified with PureLink Viral RNA/DNA Mini Kit (Invitrogen, Thermo Fisher). The EBNA1 EBV gene was amplified by real time PCR as described [54], with primers at 0.5μM. A calibrator sample used in the different real time PCR runs was analyzed

for absolute quantification of EBV genome copies (*Laboratori de Referència de Catalunya*, Barcelona). For EBV DNA quantification in re-isolated B and NK cells, DNA was purified with the Puregene Blood Core kit (Qiagen). EBNA1 gene was amplified as described above and the endogenous gene 36B4, amplified as published [55], was used as reference control for relative quantification.

To assess EBV infection of NK cells as well as control B cells, RNA was purified with RNeasy Minikit (Qiagen), retro-transcribed with Supercript III Reverse Transcriptase (Invitrogen, Thermo Fisher) and expression of BZLF1, BALF2, EBNA2 and EBNA1 genes was done as described [50] by nested PCR, except for the EBNA1 gene in which only the PCR with the second primers pair was performed. β2-microglobulin expression was measured as internal control with the following primers: Forward 5' TTAGCTGTGCTCGCGCTACTCT 3', Reverse 5' TGGTTCACACGGCAGGCATACT 3'. Real time PCR reactions were performed with Light Cycler 480 SYBRGreen I Master in a Light Cycler 480 II thermal cycler (Roche).

### Microscopy

B cells were incubated with PKH26-stained B95.8 VP for 1h at 4˚C and washed. NK cells were stained with 0.6μM CFSE. B and NK cells were mixed at 1:1 ratio with 3% EBV S- or S+ serum, spun at 230g for 1min and cultured at 37˚C for 15min, 1h or 4h. Subsequently samples were passed onto poly-Lysine-coated coverslips, spun again at 230g for 1min, left to adhere for 15min at 37˚C, fixed with 4% paraformaldehyde PBS for 15min at RT and washed with PBS. Subsequently they were: a) blocked with 1% BSA and 10μg/ml aggregated human IgG in PBS for 30 min at RT, stained with rabbit anti-human Rab5 pAb (Abcam) for 1h at RT followed by goat anti-rabbit-Alexa647 pAb (Abcam) diluted in the same buffer for 1h at RT, or b) blocked with 10% FBS 10μg/ml aggregated human IgG PBS for 30' at RT and stained with anti-CD107α-APC (Clone H4A3, BD Fastimmune) diluted in the same buffer for 1h at RT. Coverslips were then washed with PBS and were mounted with DAKO Fluorescent Mounting Media. Images (1024x1024pixels) of the green (NK; CFSE), red (VP; PKH26), far red (endosomes/lysosomes; Alexa-647/ APC), and transmitted light channels were acquired on a Leica SP8 laser scanning confocal microscope with a 63x oil immersion lens (NA 1.4), acquiring Z-slices every 0.6μm.

### Image analysis

Microscopy images were analyzed with ImageJ software (FIJI). Transmitted light images were used to delineate regions of interest (ROI) corresponding to NK (green) or B cells. For every image, automated thresholds were applied for the red and far red channels. For every ROI, total PKH26 fluorescence (Integrated Density) above the threshold of all the Z-slices was summed to obtain the total PKH26 fluorescence per cell. For every ROI, the areas of the positive signal for endosomes/lysosomes of all the Z-slices were summed to obtain the volume of the signal ($V_{e/l}$). The total area of every ROI was multiplied by the number of Z-slices to obtain the volume of the ROI ($V_{ROI}$). The VP signal colocalizing with endosome/lysosome signal was quantified with Manders coefficient ($M_{VP}$) with the JACoP Plugin [56], using the same automatic thresholds as before, and were corrected with the following formula as described [57]: $M_{VP}$ diff = $M_{VP}$—($V_{e/l}$ /$V_{ROI}$). The Pearson coefficient (R) of the red and far red intensities of pixels above Costes automatic thresholds were also calculated with the JACoP Plugin.

### Video microscopy

NK and B cells were prepared for co-culture as in confocal microscopy experiments. 15,000 NK cells and B cells were mixed in 30μl together with 3% of EBV S- or S+ sera in poly-L-lysine-coated wells of μ-slide 18 well-flat plates (Ibidi), and allowed to settle for 5 min. Cells

were then subjected to a time-lapse study in a confocal microscope. Automated multiposition live cell imaging was carried out using a 63x oil immersion objective lens (NA 1.4) and pinhole set at 1.5 Airy units in a Zeiss LSM 880 confocal microscope equipped with Predictive Focus to keep specimen in focus and an incubation system with temperature (37˚C) and $CO_2$ control (5% $CO_2$). Images of CFSE (green, NK cells) and PKH26 (red, VP) fluorophores were acquired sequentially line by line using 488 and 561 lasers lines and detection ranges at 500–550 and 570–650 nm respectively. Simultaneously, bright field images were acquired. Images of the same channel were acquired every 18 sec during 1h 11 min. Image analysis was performed using ImageJ software (FIJI).

## Data plotting and statistical analysis

Data were plotted and analyzed with GraphPad Prism 5.0. Bar graphs represent Mean-+ SEM. Shapiro-Wilk test was used to assess normal distribution. One-way ANOVA test with Bonferroni's multiple comparison test were applied for comparison of unpaired data of multiple groups following a normal distribution. Repeated measures mixed model ANOVA with Bonferroni's multiple comparison test was applied to analyze paired data grouped according to two variables. Friedman test with Dunn's post-test were applied for comparison of paired data of multiple groups. Wilcoxon and Mann-Whitney U tests were respectively applied for comparison of paired and unpaired data between two groups. Differences were considered significant at a two-sided level of <0.05. * $p < 0.05$, ** $p < 0.01$, *** $p < 0.001$.

## Supporting information

**S1 Fig. Representative gating strategy for B cells in B-NK cell co-cultures.**
(PDF)

**S2 Fig. Serum and purified serum IgG from EBV+ donors promotes NK cell-mediated elimination of viral particles coating B cells.**
(PDF)

**S3 Fig. Peripheral blood monocytes remove VP attached to B cells through anti-EBV Abs.**
(PDF)

**S4 Fig. Undetectable EBV infection in NK cells co-cultured with B95.8-coated B cells and anti-EBV Abs.**
(PDF)

**S5 Fig. CD21 acquired by NK cells through Ab-induced trogocytosis can interact with EBV VP.**
(PDF)

**S1 Video. NK cell interaction with VP-coated B cells in the presence of EBV S+.**
(AVI)

**S2 Video. NK cell interaction with VP-coated B cells in the presence of EBV S-.**
(AVI)

## Acknowledgments

We acknowledge the Advanced Light Microscopy Unit of the Center for Genomic Regulation for technical support in microscopy image acquisition and analysis, the Advanced Optical Microscopy service of the University of Barcelona, Campus Casanova, for technical support in

video microscopy, and the Flow Cytometry Core Facility of Barcelona Biomedical Research Park (PRBB). We thank Mireia Llop from the Hospital del Mar Medical Research Institute for donor blood extraction; Gemma Heredia, Araceli Cuelliga Aranda, Dr. Gemma Pérez Vilaró and Dr. Marc Talló from the Pompeu Fabra University for technical help; Dr. Giuliana Magri from the Hospital del Mar Medical Research Institute (IMIM) for reagents and advice in B95.8 culture and SN preparation.

## Author Contributions

**Conceptualization:** Elisenda Alari-Pahissa, Miguel López-Botet.

**Data curation:** Elisenda Alari-Pahissa.

**Formal analysis:** Elisenda Alari-Pahissa.

**Funding acquisition:** Miguel López-Botet.

**Investigation:** Elisenda Alari-Pahissa, Michelle Ataya, Ilias Moraitis, Miriam Campos-Ruiz, Mireia Altadill, Anna Moles.

**Methodology:** Elisenda Alari-Pahissa, Michelle Ataya, Aura Muntasell, Anna Moles.

**Project administration:** Miguel López-Botet.

**Resources:** Miguel López-Botet.

**Software:** Elisenda Alari-Pahissa.

**Supervision:** Elisenda Alari-Pahissa, Miguel López-Botet.

**Validation:** Elisenda Alari-Pahissa, Miguel López-Botet.

**Visualization:** Elisenda Alari-Pahissa, Miguel López-Botet.

**Writing – original draft:** Elisenda Alari-Pahissa, Miguel López-Botet.

**Writing – review & editing:** Elisenda Alari-Pahissa, Michelle Ataya, Mireia Altadill, Aura Muntasell, Anna Moles, Miguel López-Botet.

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
