## [Decision Letter · Decision Letter 0]

2 Mar 2021

Dear Prof. López-Botet,

Thank you very much for submitting your manuscript "NK cells eliminate Epstein-Barr virus bound to B cells through a specific antibody-mediated uptake" for consideration at PLOS Pathogens. As with all papers reviewed by the journal, your manuscript was reviewed by members of the editorial board and by several independent reviewers. In light of the reviews (below this email), we would like to invite the resubmission of a significantly-revised version that takes into account the reviewers' comments.

We cannot make any decision about publication until we have seen the revised manuscript and your response to the reviewers' comments. Your revised manuscript is also likely to be sent to reviewers for further evaluation.

Sincerely,

Erik K Flemington

Section Editor

PLOS Pathogens

Erik Flemington

Section Editor

PLOS Pathogens

Kasturi Haldar

Editor-in-Chief

PLOS Pathogens

orcid.org/0000-0001-5065-158X

Michael Malim

Editor-in-Chief

PLOS Pathogens

orcid.org/0000-0002-7699-2064

Reviewer's Responses to Questions

**Part I - Summary**

Reviewer #1: The authors demonstrate that natural killer (NK) cells prevent B cell infection by the Epstein Barr virus (EBV) in the presence of anti-viral antibody containing serum that does not inhibit infection on its own. In part this seems due to uptake of the viral particles by NK cells into early and then late endosomes. In the process some B cell markers are also transferred by trogocytosis, including CD21 the receptor for EBV attachment to B cells. The authors speculate that NK cells might limit EBV infection of B cells by non-lytic means in submucosal secondary lymphoid tissues like tonsils.

This is an interesting proposition, but it remains unclear if this is a feature of all Fc receptor carrying cells and if it is somehow involved in NK cell infection by EBV.

Reviewer #2: This paper demonstrates antibody dependent uptake of EBV viral particles in to NK cells after direct interaction with infected B cells. A number of different methods, including detection of viral gp350, GFP tagged particles, viral DNA and PKH26 labelled virus particles to detect transfer of EBV components from B cells and uptake by NK cell. The authors also demonstrate a reciprocal loss of EBV components from donor B cells. The experiments are mostly well controlled, the data is well analysed and presented and the conclusions are largely supported by the data. This study is novel as it potentially demonstrate a novel mechanism whereby NK cells could limit EBV infection in host B cells.

**Part II – Major Issues: Key Experiments Required for Acceptance**

Reviewer #1: 1. The authors suggest that the observed mechanism might be a new effector function of NK cells, but it remains unclear, if similar effects could be observed for all Fc receptor expressing cells. Are NK cells better than monocytes in removing EBV particles from B cell surfaces? NK cells should be compared in this respect with some classical phagocyte population.

2. Chronic active EBV infection (CAEBV) often spreads to other lymphocytes than B cells including NK cells. Could the observed mechanism facilitate NK cell infection by EBV? Any sign of latent EBV gene expression in NK cells after the antibody mediated uptake? As an alternative to latent EBV gene expression, GFP expression from recombinant EBV could be monitored in NK cells that have taken up EBV particles.

3. The authors observed CD19, CD20 and CD21 transfer to NK cells by trogocytosis. Is this of any functional consequence? Can CD21 expression and MHC class II up-regulation on activated NK cells be used for EBV infection? Is such infection blocked by antibodies against these respective surface markers?

Reviewer #2: 1. For flow cytometric analysis (Figures 2, 5, 6, 7 ) the authors should demonstrate with some rigour (Other than comparison with the Rituximab control) how they can exclude that a small proportion of transient B-NK cell conjugates are not being detected within the NK cell gate thereby detecting B cell markers CD19, CD20 and CD21 and also virus gp350 labelled B cells within this gate.

2. How are small populations of dead cells being excluded from the co-culture systems?

3. Fig 7. Some clarification is needed for the data in figure 7A and 7B. In 7A, the right hand plot suggests a gradual attrition of virus from B cells with or without immune sera in the absence of NK cells. Does the co-culture system not simply reflect an acceleration of the process? The attrition of S+ in the absence of NK cells implies that this process is, at least in part, passive. Significance values should also be added to these plots to compare values across the time frame and between S- and S+ data points.

4. Figure 7D and E and supplementary video microscopy data. Although the this data does support uptake of viral components, what is the cumulative frequency of PKH26+ NK cells detected in these systems at any given point across the entire time frame and how does this compare to the steady state levels detected by flow cytometry (See point 1, above).

5. The authors demonstrate the co- localisation of EBV associated components with NK cell endosomal and lysosomal markers. Can they exclude the persistence of viable viral particles?

This point should also be discussed in the context of evidence for or against a role for antibody facilitating active infection of NK cells.

**Part III – Minor Issues: Editorial and Data Presentation Modifications**

Reviewer #1: 1. The authors suggest that uncontrolled EBV infection in CD16A deficient patients points towards the importance of the described mechanism for EBV specific immune control in vivo. However, they should acknowledge that it was proposed that CD16A is required for CD2 function on NK cells outside of ADCC.

Reviewer #2: 1. Fig 6. Could this be re-ordered to separate the subplots according to experimental conditions and to label the culture conditions more clearly.

PLOS authors have the option to publish the peer review history of their article (what does this mean?). If published, this will include your full peer review and any attached files.

Reviewer #1: No

Reviewer #2: **Yes: **Martin R Goodier
---

## [Editor Report · Decision Letter 1]

4 Aug 2021

Dear Dr López-Botet

We are pleased to inform you that your manuscript 'NK cells eliminate Epstein-Barr virus bound to B cells through a specific antibody-mediated uptake' has been provisionally accepted for publication in PLOS Pathogens.

Best regards,

Claire Shannon-Lowe

Guest Editor

PLOS Pathogens

Erik Flemington

Section Editor

PLOS Pathogens

Kasturi Haldar

Editor-in-Chief

PLOS Pathogens

orcid.org/0000-0001-5065-158X

Michael Malim

Editor-in-Chief

PLOS Pathogens

orcid.org/0000-0002-7699-2064
---

## [Editor Report · Acceptance letter]

16 Aug 2021

Dear Prof. López-Botet,

We are delighted to inform you that your manuscript, "NK cells eliminate Epstein-Barr virus bound to B cells through a specific antibody-mediated uptake," has been formally accepted for publication in PLOS Pathogens.

Best regards,

Kasturi Haldar

Editor-in-Chief

PLOS Pathogens

orcid.org/0000-0001-5065-158X

Michael Malim

Editor-in-Chief

PLOS Pathogens

orcid.org/0000-0002-7699-2064